



# Marine gas-phase sulfur emissions during an induced phytoplankton bloom

Delaney B. Kilgour[1], Gordon A. Novak[1], Jon S. Sauer[2], Alexia N. Moore[2], Julie Dinasquet[3], Sarah Amiri[2], Emily B. Franklin[4], Kathryn Mayer[2], Margaux Winter[5], Clare K. Morris[3], Tyler Price[2], Francesca Malfatti[3,6], Daniel R. Crocker[2], Christopher Lee[3], Christopher D. Cappa[7], Allen H. Goldstein[4,8], Kimberly A. Prather[2,3], and Timothy H. Bertram[1]

[1]Department of Chemistry, University of Wisconsin – Madison, Madison, WI 53706, USA
[2]Department of Chemistry and Biochemistry, University of California, San Diego, La Jolla, CA 92093, USA
[3]Scripps Institution of Oceanography, University of California, San Diego, La Jolla, CA 92037, USA
[4]Department of Civil and Environmental Engineering, University of California, Berkeley, CA 94720, USA
[5]Department of Chemistry and Chemical Biology, Harvard University, Cambridge, MA 02138, USA
[6]Department of Life Sciences, Universita' degli Studi di Trieste, Trieste, Italy 34127
[7]Department of Civil and Environmental Engineering, University of California, Davis, CA 95616, USA
[8]Department of Environmental Science, Policy and Management, University of California, Berkeley, CA 94720, USA

*Correspondence to*: Timothy H. Bertram (timothy.bertram@wisc.edu)

**Abstract.** The oxidation of dimethyl sulfide (DMS; $CH_3SCH_3$), emitted from the surface ocean, contributes to the formation of Aitken mode particles and their growth to cloud condensation nuclei (CCN) sizes in remote marine environments. It is not clear whether other, less commonly measured marine-derived, sulfur-containing gases share similar dynamics to DMS and contribute to secondary marine aerosol formation. Here, we present measurements of gas-phase volatile organosulfur molecules taken with a Vocus proton transfer reaction high resolution time-of-flight mass spectrometer during a mesocosm phytoplankton bloom experiment using coastal seawater. We show that DMS, methanethiol (MeSH; $CH_3SH$), and benzothiazole ($C_7H_5NS$) account for on average over 90% of total gas-phase sulfur emissions, with non-DMS sulfur sources representing $36.8 \pm 7.7\%$ of sulfur emissions during the first nine days of the experiment in the pre-bloom phase prior to major biological growth, before declining to $14.5 \pm 6.0\%$ in the latter half of the experiment when DMS dominates during the bloom and decay phases. The molar ratio of DMS to MeSH during the pre-bloom phase (DMS:MeSH = $4.60 \pm 0.93$) was consistent with the range of previously calculated ambient DMS to MeSH sea-to-air flux ratios. As the experiment progressed, the DMS to MeSH emission ratio increased significantly, reaching $31.8 \pm 18.7$ during the bloom and decay. Measurements of dimethylsulfoniopropionate (DMSP), heterotrophic bacteria, and enzyme activity in the seawater suggest the DMS:MeSH ratio is a sensitive indicator of the bacterial sulfur demand and the composition and magnitude of available sulfur sources in seawater. The evolving DMS:MeSH ratio and the emission of a new aerosol precursor gas, benzothiazole, have important implications for secondary sulfate formation pathways in coastal marine environments.





## 1 Introduction

The ocean accounts for the largest natural source of sulfur to the atmosphere, primarily as dimethyl sulfide (DMS;

$CH_3SCH_3$) (Andreae, 1990; Simó, 2001). Current estimates for oceanic DMS emissions range between 17.6–34.4 Tg S yr$^{-1}$ (Lana et al., 2011), compared to anthropogenic DMS emission estimates of 2.20 Tg S yr$^{-1}$ (Lee and Brimblecombe, 2016). DMS has been shown to impact the production rate of secondary marine aerosol (SMA), the concentration of cloud condensation nuclei (CCN), and Earth's radiation budget by altering cloud properties (Bates et al., 1992; Carpenter et al., 2012; Charlson et al., 1987; Lana et al., 2011).


DMS is primarily produced in seawater following the bacterial cleavage of the algal metabolite dimethylsulfoniopropionate (DMSP) (Challenger and Simpson, 1948). DMSP is present in both particulate ($DMSP_p$) and dissolved ($DMSP_d$) forms, where $DMSP_p$ consists of phytoplankton intracellular DMSP, and $DMSP_d$ consists of the dissolved pool in the seawater (Kiene and Linn, 2000a). $DMSP_p$ concentrations in coastal seawater span a large range, from 5 to >300 nM, dependent on

bloom dynamics, whereas $DMSP_d$ is often present in lower concentrations (1-25 nM) and has a turnover rate of 1-129 nM d$^{-1}$ (Kiene et al., 2000; Kiene and Linn, 2000a). During blooms of DMSP-rich phytoplankton and in some colder waters, total DMSP ($DMSP_t$; $DMSP_d$ + $DMSP_p$) can significantly exceed these ranges (Kiene et al., 2019; Kiene and Linn, 2000a; Kwint and Kramer, 1996). Isotopic labeling experiments using the $^{35}S$ isotope show wide variability in the DMS yield from $DMSP_d$ (3-50%) (Carpenter et al., 2012), but the yield is commonly estimated as 10% (Kiene and Linn, 2000a). This results in

waterside DMS concentrations in the range of 1–7 nM globally, with higher values in the summer and in bloom conditions (Kiene et al., 2000; Kiene and Linn, 2000b; Lana et al., 2011). Once produced, DMS in seawater can be transformed by bacterial or photochemical processes, or converted to non-volatile sulfur, resulting in a DMS lifetime on the order of a few days in seawater (Flöck and Andreae, 1996; Kiene et al., 2000; Kiene and Linn, 2000b; Lawson et al., 2020). Approximately 10% of the dissolved DMS ventilates to the atmosphere, where it is oxidized by the hydroxyl radical (OH), halogen radicals

(Cl and BrO), and nitrate radical ($NO_3$) to form lower volatility products, including sulfur dioxide ($SO_2$) with yields ranging between 30% and 100%, and methanesulfonic acid (MSA) (Carpenter et al., 2012; Chen et al., 2018; Faloona, 2009; Lawson et al., 2020). Atmospheric $SO_2$ is further oxidized to sulfuric acid ($H_2SO_4$) and sulfate ($SO_4^{2-}$), which can lead to new particle formation, while MSA primarily contributes to particle growth (Carpenter et al., 2012).

Methanethiol (MeSH; $CH_3SH$) has also been observed in marine environments, although there are fewer measurements compared to DMS and the MeSH emission rate is thought to be a small fraction of the DMS emission rate. MeSH is the major $DMSP_d$ product (~75% yield) and is formed when bacteria demethylate or demethiolate $DMSP_d$ (Kiene, 1996; Kiene and Linn, 2000b). However, existing measurements of dissolved MeSH concentrations are significantly smaller than collocated dissolved DMS concentrations (<1.8 nM versus <6 nM) (Kettle et al., 2001). This is a result of its reaction with

dissolved organic matter and its rapid incorporation into bacterial cells where it is used to form methionine (Flöck and



Andreae, 1996; Kiene, 1996; Kiene et al., 1999). This leads to a dissolved MeSH lifetime on the order of minutes to an hour, which is considerably shorter than that of DMS (Lawson et al., 2020). Although the fraction of MeSH that ventilates to the atmosphere is poorly constrained, it serves as an additional source of reduced sulfur to the marine atmosphere and has a faster reaction rate with OH at 298 K ($3.3 \times 10^{-11}$ $cm^3$ $molecules^{-1}s^{-1}$) compared to that of DMS with OH at 298 K ($4.8 \times 10^{-12}$

$cm^3$ $molecules^{-1}s^{-1}$ via H-abstraction and $1.7 \times 10^{-12}$ $cm^3$ $molecules^{-1}s^{-1}$ via OH addition) (Atkinson et al., 2004), suggesting MeSH could also contribute to marine boundary layer (MBL) $SO_2$ and sulfate aerosol.

The emission ratio of DMS to MeSH (DMS:MeSH) is a sensitive indicator of $DMSP_t$ production and degradation pathways, as well as the lifetimes of DMS and MeSH in seawater. The cleavage pathway that produces DMS is in competition with the

demethylation/demethiolation pathway that produces MeSH. The favored pathway is a function of both the concentration of $DMSP_t$ and the bacterial sulfur demand (Carpenter et al., 2012; Kiene et al., 2000; Kiene and Linn, 2000a; Vila et al., 2004). Low bacterial sulfur demand and high $DMSP_t$ concentrations promote DMS production, while high bacterial sulfur demand and low $DMSP_t$ concentrations promote MeSH production (Carpenter et al., 2012; Kiene et al., 2000; Kiene and Linn, 2000a).


Fluctuations in the available sulfur pool and bacterial sulfur demand can translate to significant variability in waterside measurements of DMS:MeSH over the open ocean. In upwelling regions of the Atlantic Ocean, waterside DMS:MeSH has been shown to range between 1 and 30 (Kettle et al., 2001). Measurements made in the Baltic Sea, Kattegat/Skagerrak, and North Sea have shown waterside DMS:MeSH of 16, 20, and 6, respectively (Leck and Rodhe, 1991). More recently,

measurements in the subarctic northeast Pacific Ocean showed waterside DMS:MeSH ranged between 2–5.3 resulting in a calculated average DMS:MeSH sea-to-air flux ratio of 6 (Kiene et al., 2017). In the southwest Pacific Ocean, the reported DMS:MeSH flux ratio varied between 3 and 7 as estimated by the nighttime concentration accumulation method (Lawson et al., 2020). The Henry's law constants and diffusion constants in water at 298 K for DMS ($5.6 \times 10^{-3}$ mol $m^{-3}$ $Pa^{-1}$; $1.217 \times 10^{-5}$ $cm^2$ $s^{-1}$) and MeSH ($3.8 \times 10^{-3}$ mol $m^{-3}$ $Pa^{-1}$; $1.556 \times 10^{-5}$ $cm^2$ $s^{-1}$) are similar (Gharagheizi, 2012; Sander, 2015), implying

the dissolved concentration ratio in the seawater is directly related to the emission ratio. Periods of low DMS:MeSH suggest that MeSH could impact oxidative capacity of the MBL by providing a significant source of reduced sulfur to the atmosphere.

Other sulfur species, including dimethyl disulfide, carbon disulfide, and carbonyl sulfide, have previously been measured in

the seawater in highly productive regions, though in significantly smaller quantities than DMS (Kettle et al., 2001; Leck and Rodhe, 1991). Recently, a previously unobserved biogenic marine volatile sulfur molecule, methane sulfonamide, was measured in the gas-phase near an upwelling region of the Arabian Sea at mixing ratios up to 33% of DMS (Edtbauer et al., 2020). Additionally, the recent discovery of the DMS oxidation product hydroperoxymethyl thioformate (HPMTF) has prompted researchers to reexamine our understanding of the sulfur cycle (Veres et al., 2020). The combination of these





findings raises questions regarding whether organosulfur molecules emitted in smaller quantities than DMS are important to the sulfur budget and contribute to sulfate aerosol and CCN in the marine atmosphere.

Here we report measurements of gas-phase volatile organosulfur molecules, with specific focus on DMS, MeSH, and a marine sulfur-containing molecule not reported prior to this experiment, benzothiazole (Franklin et al., 2021). These measurements were made during a mesocosm bloom experiment in a low-oxidant wave channel at the Scripps Institution of Oceanography in La Jolla, California. We examine how the distribution of emitted gas-phase sulfur molecules evolves as a function of bloom stage and provide insight into biological and environmental controls on the production and loss processes of these gases.

## 2 Methods

### 2.1 Scripps Institution of Oceanography Wave Channel and Mesocosm Experiment

The experiment was conducted at the Scripps Institution of Oceanography Hydraulics Laboratory wave channel as part of the Center for Aerosol Impacts on Chemistry of the Environment's intensive Sea Spray Chemistry And Particle Evolution (SeaSCAPE) experiment in July and August 2019. The collaborative study aimed to determine the impacts of biological activity, oxidative aging, and photochemistry on the emission of marine trace gases, the production of nascent sea spray aerosol, and the composition of secondary marine aerosol. Here, we present analysis of gas-phase sulfur species from the third of three phytoplankton blooms. This part of the mesocosm experiment lasted 21 days, where day 0 marks the time when the wave channel was filled with seawater from the Pacific Ocean, pumped directly from below Ellen Browning Scripps Memorial Pier (herein Scripps Pier) in La Jolla, CA. Details of the wave channel setup and wave-breaking mechanism have been described elsewhere (Prather et al., 2013) and additional detail on this collaborative study is provided in the supplementary information (S1). A phytoplankton bloom was induced through a series of f/2 and f/20 growth medium and silicate additions. Details and timing of nutrient additions and perturbations to the mesocosm system are in the supplementary information (S2 and Table S1).

Due to the large volume of the wave channel, it is challenging to fully clean the headspace of background trace gases. As a result, all gas-phase measurements were made from an isolated sampling vessel (ISV) (Fig. S1) that circulated wave channel seawater using a peristaltic pump, providing a water residence time of 29 minutes. Its headspace was continuously purged with zero air (air residence time of 5 minutes) from a zero air generator (Sabio 1001), providing a headspace with low NOx, $O_3$, and background VOC. For this work, the ISV was sampled at 100 standard cubic centimeters per minute (sccm) through an approximately 2.5 m 0.25" O.D. PFA tube. While the ISV and wave channel were illuminated with fluorescent lights during gas-phase measurements, these do not mimic the solar spectrum reaching the ocean's surface, providing a key difference between this work and referenced work studying gaseous emissions in the ambient environment.



## 2.2 PTR-ToF-MS Measurements of Reduced Sulfur Compounds

A Vocus proton transfer reaction time-of-flight mass spectrometer (PTR-ToF-MS) (TOFWERK, Aerodyne, Inc.) was deployed to measure gas-phase volatile sulfur molecules. The Vocus instrument has a high resolving power ($m/\Delta m$ >5000)
and 1–2 orders of magnitude improved sensitivity over prior low-pressure PTR-ToF-MS instruments, allowing detection of the sub-ppt level gases observed in this study (Krechmer et al., 2018).

Mass spectra were collected from 19–500 m/$Q$ and saved at 1 Hz time resolution. Peak fitting and integration were completed in Tofware v3.1.2 (TOFWERK). The Vocus instrument parameters used in the study are as follows: The big
segmented quadrupole (BSQ) voltage was 275 V, acting as a high-pass band filter to reduce the ion transmission of low mass (<35 m/$Q$) ions (Krechmer et al., 2018). The focusing ion-molecule reactor (FIMR) was operated at a high reduced field strength (E/N = 143 Td) with a pressure of 1.5 mbar, axial electric field gradient of 41.5 V cm$^{-1}$, and was heated to 100 °C. The high reduced field strength lessened reagent ion clustering and increased fragmentation of some ions.

Measurements of the ISV headspace were taken for approximately one hour at 9 am and one hour at 2 pm each day, and daily averages were calculated as the average over the total two-hour measurement period. Instrument background signals were determined approximately 8 times daily by overflowing the Vocus inlet with zero air from the same zero air generator that provided air to the ISV headspace. Daily average background signals were used for background correction. Calibration factors for DMS and MeSH were determined by diluting a gas standard (5.08 ppm ± 5% DMS, Praxair; 6.111 ppm ± 5%
MeSH, Airgas) into zero air. The benzothiazole calibration factor was measured using a syringe pump to inject dilute solutions of benzothiazole (96%, Sigma-Aldrich) in cyclohexane (99.5%, Sigma-Aldrich) into zero air carrier gas flow. The dry sensitivities to DMS, MeSH, and benzothiazole are 3.0, 1.0, and 5.8 cps ppt$^{-1}$, respectively. Other sulfur-containing species (listed in Table S2) were quantified using the DMS sensitivity, as the proton transfer rate constant for DMS is similar to the proton transfer rate constants for other sulfur-containing species (Sekimoto et al., 2017). All molecules were identified
and quantified by their protonated ion (MH$^+$). Identifications of non-calibrated large mass species (>100 g mol$^{-1}$) in Table S2 were provided by thermal desorption two-dimensional gas chromatography coupled with electron ionization time-of-flight mass spectrometry (TD-GCxGC-ToF-MS) (Franklin et al., 2021).

## 2.3 Waterside Measurements at Wave Channel

The following waterside variables were measured continuously for indication of bloom progression: fluorescent dissolved
organic matter (FDOM) and chlorophyll-a (ECO-Triplet-BBFL2; Sea-Bird Scientific), dissolved oxygen and water temperature (SBE 63 Optical Dissolved Oxygen Sensor; Sea-Bird Scientific), and salinity (SBE 37 SI MicroCAT; Sea-Bird Scientific).





Bulk water samples collected from the wave channel were used for daily measurements of the following: heterotrophic
bacteria abundance measured with flow cytometry (Gasol and Del Giorgio, 2000), bacterial productivity determined by
radiolabeled leucine incorporation (Kirchman et al., 1985; Azam and Smith, 1992; Simon and Azam, 1989), phytoplankton
enumeration determined by the Utermöhl method (Edler and Elbrächter, 2010) and dissolved DMS, $DMSP_p$, and $DMSP_d$
measured by a home-built purge and trap system (Wurl, 2009) coupled to a chemical ionization mass spectrometer with
benzene cluster cation reagent ions (Fig. S2) (Lavi et al., 2018; Kim et al., 2016). More information on these methods is
described in the supplementary information (S3 and S4).

## 3 Results and Discussion

### 3.1 Vocus PTR-ToF-MS Characterization of Organosulfur Molecules

Krechmer et al. (2018) previously characterized the Vocus performance in a lab setting and Li et al. (2020b, a) have
described its abilities in forest sites. What follows is the first description of the instrument's capabilities for studying marine
trace gases, which comprise a unique subset of VOCs that are often emitted in smaller quantities than in forest or urban
environments. In this manuscript we focus on organosulfur molecules.

Twenty-eight sulfur-containing ions were detected in the mass spectrum (Table S2). In addition to ions corresponding to the
molecules DMS, MeSH, and benzothiazole, these included ions such as $C_3H_6SH^+$, $C_2H_6S_2H^+$, $C_4H_8SO_2H^+$, $C_{10}H_{16}SH^+$, and
$C_{11}H_{16}SH^+$. A sample mass spectrum highlighting sulfur-containing ions and the high-resolution fit around DMS is in Fig. 1.
Several ions are detected at the unit masses of important marine gases (Fig. 1b), with ions at m/Q 49 (the m/Q corresponding
to MeSH) including $CClHH^+$, $HO_3^+$, and $CH_4O_2H^+$, and ions at m/Q 63 (the m/Q corresponding to DMS) including
$H_2CO_3H^+$, $H_2N_2O_2H^+$, and $C_2H_6O_2H^+$. Previous open ocean measurements of DMS and MeSH have been reported at unit
mass m/Q 63 and m/Q 49, respectively (Lawson et al., 2020). In this work with coastal seawater in an indoor laboratory
setting, MeSH constituted 73.9 ± 12.9% of m/Q 49 and DMS constituted 76.8 ± 18.0% of m/Q 63. Therefore, the high
resolution of the Vocus ensured accurate quantitative measurements of DMS and MeSH.





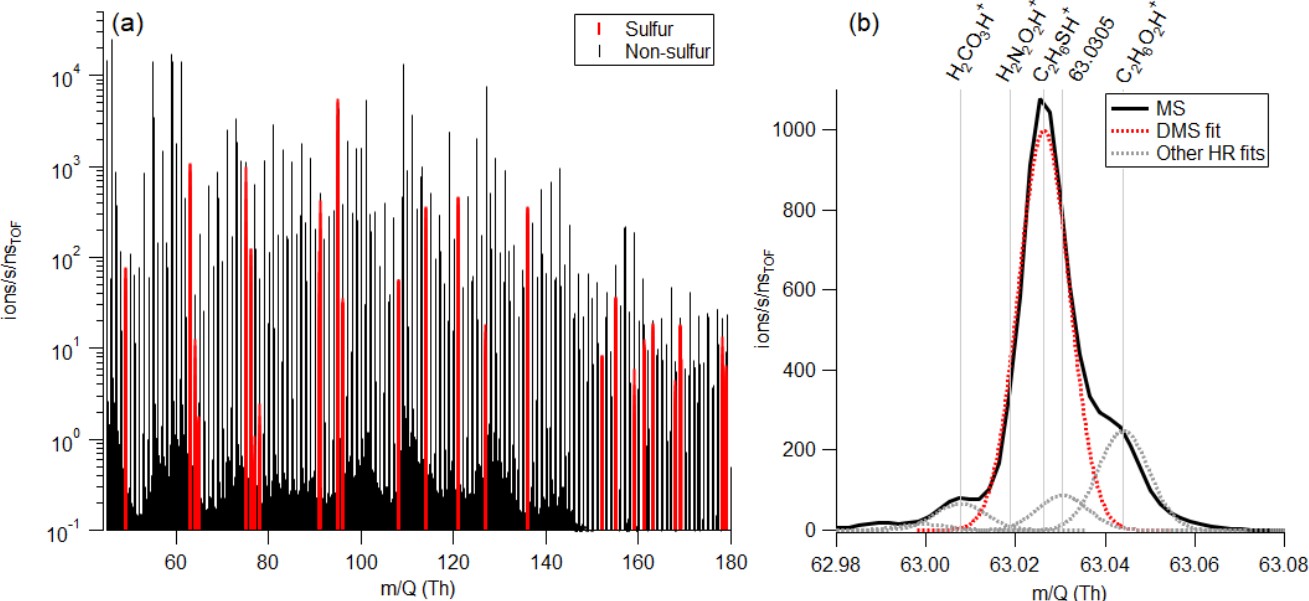

**Figure 1: (a) Sample mass spectrum corresponding to an ISV headspace measurement. Peaks highlighted in red contain sulfur. (b) High-resolution fit around DMS ($C_2H_6SH^+$). For this mass spectrum, DMS makes up 72.2% of the total ion current at m/Q 63 with $C_2H_6O_2H^+$ being the second largest peak.**

In Fig. S3 and Fig. S4, we show calibration curves for DMS, MeSH, and benzothiazole, and observed fragments of these molecules. Limits of detection for DMS, MeSH, and benzothiazole at 1 minute averaging time were $0.20 \pm 0.49$ ppt, $1.5 \pm 0.25$ ppt, and $0.42 \pm 0.14$ ppt, respectively. While observations of DMS, MeSH, and benzothiazole in this study were well above the instrument's limits of detection, open ocean measurements of DMS and MeSH in non-bloom conditions are on the order of tens of ppt, making the Vocus with its low limits of detection an ideal instrument to use in such conditions.

Krechmer et al. (2018) demonstrated the Vocus sensitivity to a number of non-sulfur containing VOCs is independent of relative humidity due to the high water vapor mixing ratio in the focusing ion-molecule reactor causing the relative humidity in the small volume of sample air to have limited effects on ion-molecule reactions in the Vocus. In Fig. 2 we show the sensitivity to DMS is humidity-independent, while a small humidity-dependence exists for MeSH between $2.7 \times 10^{-4}$ kg m$^{-3}$ and 0.018 kg m$^{-3}$. The upper half of these values (0.009–0.018 kg m$^{-3}$), corresponding to roughly 40–80% relative humidity at 25 °C, are commonly observed over mid-latitude oceans (Liu et al., 1991). DMS signal was within the standard deviation for all absolute humidity values tested, but MeSH signal was strongly anticorrelated with absolute humidity ($R^2 = 0.96$) and decreased 20% across the absolute humidity range tested.



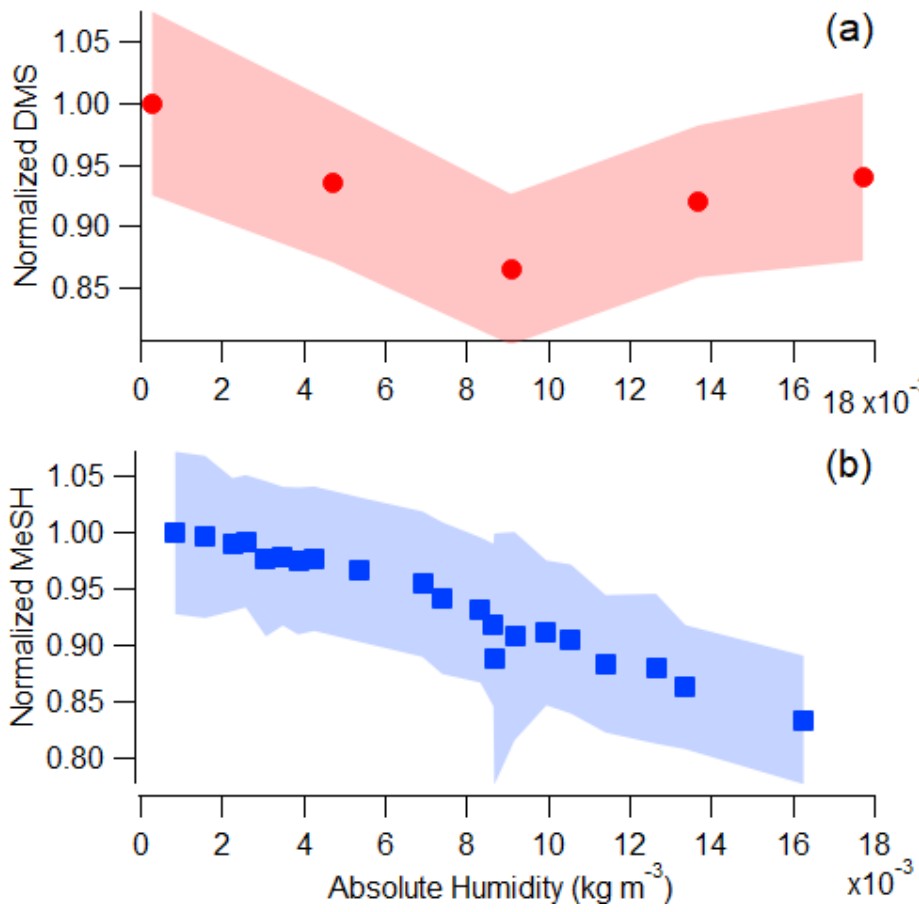

**Figure 2: Vocus PTR-ToF-MS signal as a function of absolute humidity for a constant flow of (a) DMS (measured at m/Q 63.0263) and (b) MeSH (measured at m/Q 49.0107) in humidified zero air. Absolute humidity values between 0.009 and 0.018 kg m⁻³ are commonly measured in the MBL in mid-latitude oceans (Liu et al., 1991).**

The decrease in MeSH with absolute humidity is likely a result of conversion on inlet surfaces in addition to humidity-dependent changes in ion chemistry. It has been well-documented in the literature that MeSH oxidation to dimethyl disulfide (DMDS; $C_2H_6S_2$) can occur on metal surfaces (Perraud et al., 2016). We used stainless steel fittings on the tubing and inlet so MeSH loss and DMDS production was a possibility to consider in our system. MeSH and DMDS (measured as $C_2H_6S_2H^+$) were tightly correlated ($R^2 = 0.98$) in MeSH calibrations (done after SeaSCAPE), with DMDS representing roughly 12% of the MeSH signal. This is likely a result of conversion in the inlet or in the calibration standard. However, there was no correlation ($R^2 = 0.017$) between DMDS and MeSH during SeaSCAPE, suggesting that the on average 7 ppt of DMDS measured during SeaSCAPE has a seawater source and cannot solely be a result of inlet conversion of MeSH (Fig. S5).





### 3.2 Gas-Phase Sulfur Budget during Bloom

Figures 3a and 3b depict the progression of the bloom in the wave channel and the effects of perturbations listed in Table S1 through waterside measurements including chlorophyll-a, heterotrophic bacteria, $DMSP_t$, and fluorescent dissolved organic matter (FDOM). Prior to the first nutrient addition on day 2, the mean chlorophyll-a concentration was $0.80 \pm 0.08$ µg L$^{-1}$,

mean heterotrophic bacteria abundance was $2.97 \times 10^9 \pm 1.27 \times 10^9$ cells L$^{-1}$, and mean $DMSP_t$ was $51.2 \pm 20.7$ nM. $DMSP_t$ and heterotrophic bacteria had a small peak on days 5 and 6, respectively, but chlorophyll-a remained low at less than 1.25 µg L$^{-1}$ until day 7. Chlorophyll-a began to rise on day 7, reaching 2.93 µg L$^{-1}$, indicative of the start of a small bloom. This in situ bloom was enhanced through the addition of a 300-gallon tank of seawater containing healthy biomass, dominated by diatoms and with chlorophyll-a measuring 43.8 µg L$^{-1}$, on day 9, causing significant responses in chlorophyll-a, $DMSP_t$, and

heterotrophic bacteria. This added seawater was collected the same way as the water in the wave channel but on a different date. It was immediately spiked with f/2 nutrients and left outside for four days until the phytoplankton bloom reached the exponential growth phase at which time it was added to the wave channel. $DMSP_t$ and chlorophyll-a peaked approximately one day after the tank addition at values of 224 nM and 25.9 µg L$^{-1}$, respectively. Heterotrophic bacteria had a small peak corresponding to the chlorophyll-a peak before reaching its maximum concentration four days after $DMSP_t$ and chlorophyll-

a at $1.3 \times 10^{10}$ cells L$^{-1}$. Chlorophyll-a concentrations were sustained at elevated values around 7 µg L$^{-1}$ following the peak, while heterotrophic bacteria measurements showed a local maximum on day 18.





**Figure 3: (a)** Time series of (a) 2-hour average chlorophyll-a and daily heterotrophic bacteria concentrations in the wave channel, **(b)** daily DMSP$_t$ and FDOM, **(c)** DMS, MeSH, benzothiazole, and the sum of other detectable sulfur ions in the mass spectrum in absolute concentration, and **(d)** fractional contribution of DMS, MeSH, benzothiazole, and the sum of other detectable sulfur ions in the mass spectrum to the total measured gas-phase sulfur budget by the Vocus. Fractional contribution is calculated by mixing ratio. The wave channel was filled on day 0 and nutrients were added on days 2.2 and 3.2. The tank of productive seawater and more nutrients were added on day 8.9, shown by the vertical line in Fig. 3. More details on these changes are recorded in Table S1.




The waterside measurements in Fig. 3a and Fig. 3b provide context for understanding the gas-phase sulfur emissions displayed in Fig. 3c and Fig. 3d. The organosulfur molecules studied are DMS, MeSH, benzothiazole, and "other S", where "other S" corresponds to the sum of 25 other detectable sulfur-containing ions in the mass spectrum. The other S signal was distributed among ions, with ions corresponding to DMDS and larger sulfur-containing compounds ($C_4H_8SO_2H^+$, $C_{10}H_{16}SH^+$, $C_{11}H_{16}SH^+$) present with appreciable signal throughout the bloom. Only three ions in the other S signal showed a correlation

of $R^2 > 0.5$ with either DMS, MeSH, or benzothiazole. Known fragments of DMS, MeSH, and benzothiazole were not included in the other S signal as they are not unique molecules. Initial concentrations of DMS, MeSH, benzothiazole, and other S in the ISV headspace at the start of the experiment when seawater was first added were 545, 97, 41, and 141 ppt, respectively. DMS and MeSH increased from the beginning of the bloom, with DMS peaking at 5690 ppt on day 13 and MeSH peaking at 274 ppt on day 11. These values are significantly higher than what is routinely measured over the open

ocean (Lawson et al., 2020; Leck and Rodhe, 1991), likely owing to multiple additions of concentrated nutrients that induced the intense phytoplankton bloom and the gas equilibration time in the ISV. Other S and benzothiazole peaked earlier in the experiment on day 7 when some other anthropogenic gases, including benzophenone and naphthalene, peaked (Franklin et al., 2021). The other S signal from this day was primarily driven by contributions from $C_4H_8SO_2H^+$, which could be indicative of the molecule sulfolane, and $C_{10}H_{16}SH^+$, and $C_{11}H_{16}SH^+$.


   Benzothiazole is a reduced sulfur molecule measured in the ISV headspace that contributed significantly to the gas-phase sulfur budget during the bloom. Since the Henry's law constant for benzothiazole in water is three to four orders of magnitude higher than that of DMS and MeSH, it is possible that the emission ratio of benzothiazole to DMS and MeSH is uniquely sensitive to the water and air flow rates in the ISV (Sander, 2015). Benzothiazole has both biological and

anthropogenic sources. It is naturally produced by the γ-Proteobacteria, *Pseudomonas fluorescens* (Le Bozec and Moody, 2009), and the Actinobacteria, *Micrococcus* sp. (Schulz-Bohm et al., 2017), both of which can be found in seawater. Benzothiazole ($C_7H_5NS$) belongs to a class of structurally similar molecules called benzothiazoles, which are a group of high production volume chemicals found in wastewater and urban runoff (De Wever and Verachtert, 1997; Hidalgo-Serrano et al., 2019). Based on analysis in Franklin et al. (2021) where the benzothiazole molecule was (1) consistently observed in

significant quantities in the dissolved, gas, and aerosol phases, (2) the gas-phase molecule displayed temporal behavior similar to a group of other anthropogenic gases, and (3) anthropogenic benzothiazole tracer species were observed, we attribute its source as primarily anthropogenic from the presence of pollutants in coastal waters. The two benzothiazole measurements, here by the Vocus PTR-ToF-MS and in Franklin et al. (2021) by TD-GCxGC-ToF-MS, differ in absolute quantities but are highly correlated ($R^2 = 0.91$) (Franklin et al., 2021). Additionally, Franklin et al. (2021) showed the

oxidation of benzothiazole in a potential aerosol mass–oxidation flow reactor produced increasing amounts of secondary aerosol and $SO_2$ from 2.9–4.7 days of equivalent aging. As a result, we suspect benzothiazole may be important in coastal regions, but is not expected to be a significant sulfur source in open ocean regions.





We find little evidence for emissions of DMS oxidation products. Dimethyl sulfoxide (DMSO) did not have a signal-to-noise
ratio above three on any day of the bloom. Dimethyl sulfone (DMSO$_2$) was present through day 6 prior to the rise in
chlorophyll-a, then decayed to below the detection limit (Fig. S6). The observed small concentrations of DMS oxidation
products relative to DMS are expected as the mesocosm experiment was conducted in a low-oxidant indoor environment.
Methane sulfonamide (MSAM) measured on average less than 1 ppt during the experiment and showed no positive
correlation with the DMS oxidation product, DMSO$_2$ (Fig. S6). This is surprising as MSAM was recently measured at
mixing ratios of up to 60 ppt corresponding to up to 33% of DMS in upwelling areas of the Arabian Sea and was correlated
with DMSO$_2$ (r = 0.8) (Edtbauer et al., 2020).

In Fig. 3c, we show that non-DMS sulfur sources comprise a larger fraction (36.8 ± 7.64%) of the gas-phase sulfur budget in
the pre-bloom period prior to the peak in chlorophyll-a on day 10. The contribution of non-DMS to total sulfur in this period
is higher than in previous measurements, such as those of DMS, MeSH, DMDS, and carbon disulfide reported by Leck and
Rodhe (1991), where waterside non-DMS/total sulfur was on average 10, 6, and 16% in the Baltic Sea, Kattegat/Skagerrak,
and North Sea, respectively, and average chlorophyll-a was 0.5–1.9 µg L$^{-1}$. When comparing the same sulfur molecules as in
Leck and Rodhe (1991), non-DMS represents 18.7 ± 3.4% of total sulfur for the same period. This is within the range of
observations in the North Sea and suggests that this difference is either driven by a subset of sulfur molecules perhaps unique
to a coastal environment or the measurement technique. Another possible cause for the high observed non-DMS sulfur is that
the distribution and magnitude of organosulfur emissions may have been altered through the process of water collection,
transport, and wave channel filling, when the water temperature increased 4 °C during the first two days. When chlorophyll-
a was low and after the peak in bulk water heterotrophic bacterial abundance on day 15, DMS alone accounted for
approximately 87% of the sulfur budget. This is within the range of previous measurements of DMS contribution to total
sulfur made by Leck and Rodhe (1991).

### 3.3 The DMS:MeSH Molar Ratio

In the remaining sections of the paper, we focus our analysis of the gas-phase sulfur budget on DMS and MeSH, as they
collectively accounted for 84 ± 8.1% of the total sulfur budget. Further, since their production and loss are inherently linked,
measurements of the molar ratio of DMS:MeSH provides unique perspective on waterside sulfur chemistry. Given that the
Henry's law constants and diffusion constants for DMS and MeSH are roughly the same, we expect the measured molar ratio
of DMS:MeSH in the headspace to reflect the dissolved concentration ratio in the seawater. In this work, DMS and MeSH
increased at approximately the same rate from day 1 to day 9 of the bloom shown by their similar slopes in Fig. 4a. This
resulted in a relatively constant DMS:MeSH of 4.6 ± 0.9 during the first 9 days, shown in Fig. 4b. This value is consistent
with the range of previously reported waterside concentration and sea-to-air fluxes of DMS:MeSH measured over the open
ocean during non-bloom conditions (Kettle et al., 2001; Leck and Rodhe, 1991) and during a modest bloom (chlorophyll-a <
3 µg L$^{-1}$) (Lawson et al., 2020). The stable DMS:MeSH value is unexpected as it was sustained through three nutrient





additions and the addition of a tank containing highly productive seawater (Table S1). Following the peak in MeSH on day 11, DMS:MeSH began to increase considerably. The peak in DMS:MeSH on day 13 was driven by a maximum in DMS concentration relative to declining MeSH concentrations, representing a DMS increase of 3340 ppt and MeSH decrease of

150 ppt from day 11 to 13. Additionally, the period of increasing DMS:MeSH from day 10–13 likely reflects the substantially shorter waterside lifetime of MeSH compared to DMS. Maximum DMS concentrations occurred 3 days after peak DMSP$_t$, whereas maximum MeSH concentrations occurred 1 day after peak DMSP$_t$, suggesting quick turnover of DMSP$_d$ to form MeSH. The DMS:MeSH peak on day 17 was driven by low MeSH concentrations (35 ppt MeSH) relative to DMS (2490 ppt DMS). The episodic DMS:MeSH variations around days 13 and 17 are likely the results of external

perturbations to the wave channel water (Table S1) which affected water mixing and algal cell lysis. Despite these two external factors, DMS:MeSH increased significantly from the initial to final day of the experiment, reaching values higher than what have been observed in previous oceanic studies. Thus, increases in the DMS:MeSH ratio might reflect the biological dynamics induced in our controlled system.

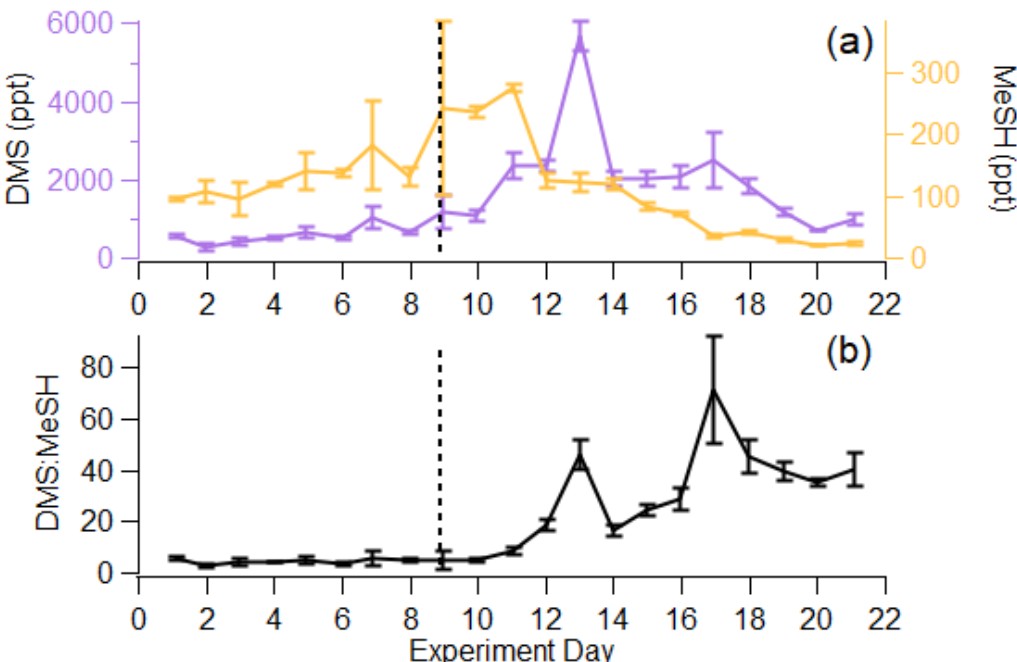

**Figure 4: (a) Time series of DMS and MeSH in the ISV, where error bars indicate the standard deviation over the 1 Hz measurements for the day. (b) Time series of DMS:MeSH, where error bars indicate the error propagation from (a). Vertical line represents the addition of highly productive seawater and more nutrients.**



### 3.3.1 Observed Correlations of DMS:MeSH with Seawater Properties

In what follows, we explore in detail the factors that control the DMS:MeSH emission ratio. Properties with the potential to impact the waterside production and loss of DMS and MeSH are examined, including chlorophyll-a, FDOM, wave channel water temperature, dissolved oxygen, and salinity. Later, we focus on a subset of these measurements that provide insight into the transformation of precursor sulfur molecules to DMS and MeSH, namely $DMSP_t$, bacterial sulfur demand, and methionine aminopeptidase activity. Regressions of these variables against DMS:MeSH are shown in Fig. 5.


Chlorophyll-a, serving as a metric for phytoplankton biomass, and thus, intracellular DMSP ($DMSP_p$) is expected to trend with total available sulfur and influence production of DMS and MeSH (Galí et al., 2015). FDOM is expected to positively correlate with DMS:MeSH due to both MeSH loss by reaction with DOM (Lawson et al., 2020) and its impact on DMS production. As the concentration and chemical complexity of FDOM increases during a bloom, the available sulfur

compounds are also expected to increase as there is a release of sulfur-rich amino acids in addition to DMSP (Pinhassi et al., 2005; Meon and Kirchman, 2001). An excess of available sulfur will favor DMS production, leading to an increase in DMS:MeSH, evidenced by the weak ($R^2 = 0.24$) positive correlation in Fig. 5b. Temperature and dissolved oxygen may influence DMS:MeSH through their relationship with bacterial growth rates and DMSP conversion (Kiene and Linn, 2000b, a). Both have weak correlations with DMS:MeSH in Fig. 5c and Fig. 5d. While the strong anticorrelation between

DMS:MeSH and salinity in this experiment is in accordance with prior experiments constraining the DMSP demethylation/demethiolation pathways as a function of salinity (Magalhães et al., 2012; Salgado et al., 2014), we argue that the observed anticorrelation in this experiment is simply a correlation and not suggestive of a salinity control. This observation is discussed further in the supplementary information (S5). The remaining sections of the paper will focus specifically on the relationship between bacterial sulfur demand, $DMSP_t$, and methionine, as this is expected to modulate the

fate of DMSP (Kiene et al., 2000).

**Figure 5: Regressions of DMS:MeSH versus waterside variables measured in the wave channel: (a) chlorophyll-a, (b) FDOM, (c) wave channel water temperature, (d) dissolved oxygen, (e) salinity, (f) DMSP$_t$, (g) bacterial sulfur demand (using a cellular C:S ratio in bacteria of 248 (Cuhel et al., 1982)), and (h) methionine aminopeptidase activity. The color of the marker indicates the day of the bloom. Panel a uses two hour averages, panels b, c, d, and e use minute averages, and panels f, g, and h use daily averages.**



### 3.3.2 Biological Influences on DMS:MeSH

DMSP is a precursor for both MeSH and DMS. $DMSP_t$ ranged from 15.3 to 224 nM (Fig. 6a), similar to concentrations observed in other phytoplankton blooms (Galí et al., 2015). Taking chlorophyll-a as a proxy for phytoplankton biomass, the ratio of $DMSP_t$:chlorophyll-a suggests that the bloom was largely dominated by low DMSP producers after the first few days
(Fig. S7), in accordance with the observed large diatom population (Fig. S8) (Dani and Loreto, 2017; McParland and Levine, 2019).

Further, assuming the lifetimes of DMS and $DMSP_t$ in the water are the same, the waterside DMS:$DMSP_t$ ratio can be used to estimate the $DMSP_t$ to DMS conversion efficiency in the seawater. While typically considered to be around 10% (Kiene
and Linn, 2000b; Lizotte et al., 2012; Vila-Costa et al., 2008), waterside DMS:$DMSP_t$ was much lower in this study, ranging between 0.38% and 8.30% (average 2.88%) (Fig. S7). This suggests there was low DMS production from DMSP, which could be a result of diatoms dominating the experiment while other taxa capable of directly producing DMS from DMSP (such as dinoflagellates and haptophytes) being less abundant (Lizotte et al., 2012; Stefels and van Boekel, 1993). Additionally, low DMS:$DMSP_t$ could be representative of a significant DMS loss either through ventilation or biotic or
abiotic transformations in the seawater.



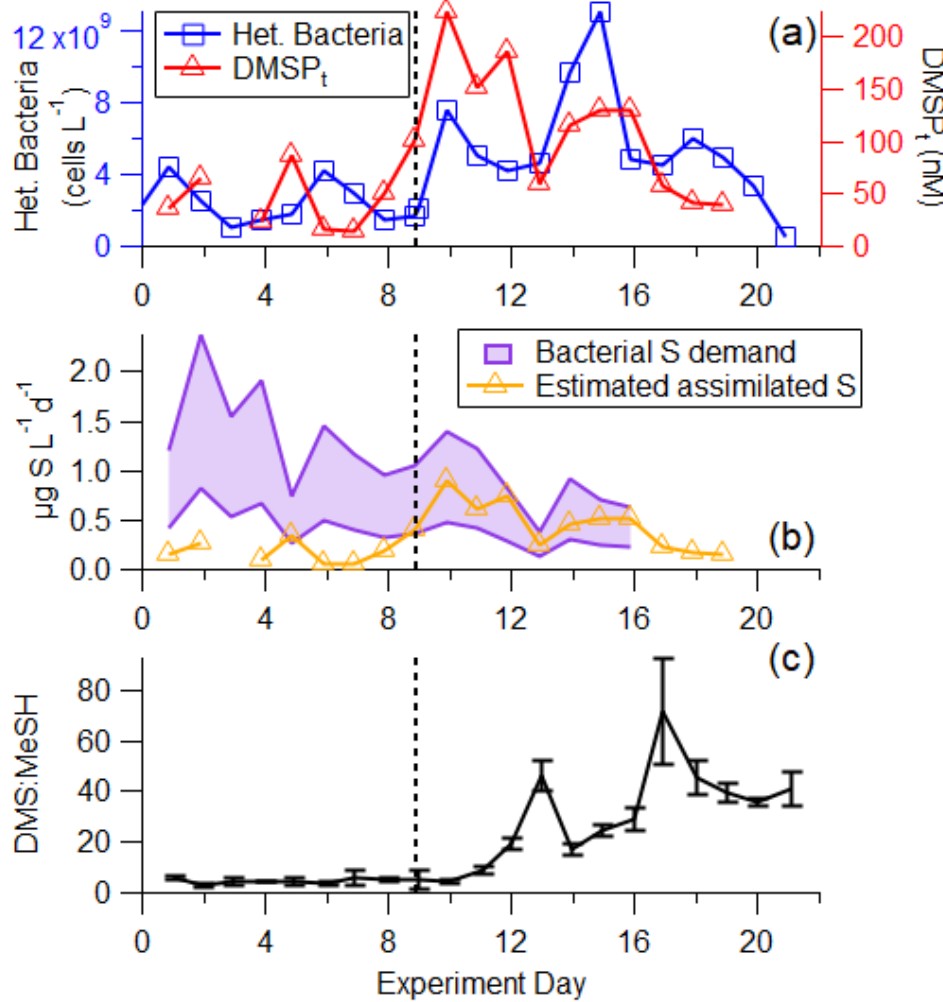

**Figure 6: (a) Time series of heterotrophic bacteria and DMSP$_t$. (b) Time series of calculated bacterial sulfur demand (cellular C:S in bacteria of 86-248) and estimated assimilated sulfur assuming 25% of DMSP$_t$ is assimilated by bacteria (Fagerbakke et al., 1996; Cuhel et al., 1982; Kiene and Linn, 2000a). (c) DMS:MeSH is low in the pre-bloom at the beginning of the experiment, before increasing significantly driven by different sources of available sulfur. Vertical line represents the addition of productive seawater and additional nutrients to the wave channel.**

The dynamics between bacterial sulfur demand and available sulfur sources are important for regulating the fate of DMSP and therefore the DMS:MeSH ratio (Kiene et al., 2000). Bacterial sulfur demand was calculated using measured bacterial productivity, assuming lower and upper limits on cellular C:S ratios in bacteria of 86 (Fagerbakke et al., 1996) and 248 (Cuhel et al., 1982; Kiene and Linn, 2000a), and assuming 25% of sulfur from DMSP$_t$ is assimilated by bacteria (Pinhassi et al., 2005). The comparison between assimilated sulfur from DMSP$_t$ and bacterial sulfur demand (Fig. 6b) shows that at the beginning of the experiment during the pre-bloom stage, bacterial sulfur demand exceeded sulfur available from DMSP, suggesting that all the DMSP was channeled toward the demethylation pathway and the formation of MeSH, and that other





sulfur sources complemented the bacterial demand. After the peak of the bloom, the assimilated sulfur from $DMSP_t$ exceeded the bacterial sulfur demand (assuming a cellular C:S ratio of 248), suggesting that bacteria produced DMS from the excess sulfur source in the latter half of the experiment, leading to a significant increase in measured DMS:MeSH (Fig.6c). This preference toward DMSP cleavage at the end of this bloom is likely due to an increase in the amount and chemical complexity of dissolved organic matter and the presence of other forms of available sulfur often observed at the

end of phytoplankton blooms (Pinhassi et al., 2005). Since $DMSP_t$ represents a small fraction of bioavailable carbon throughout the bloom (<1%), other existing sulfur sources in the carbon pool may be more easily accessible to bacteria than DMSP.

The preferential assimilation of other sulfur sources than DMSP is further supported by strong correlations between

DMS:MeSH and aminopeptidase activities (Fig. S9), particularly methionine aminopeptidase ($R^2$ = 0.82) (Fig. 5h). Aminopeptidases catalyze cleavage of amino acids from proteins and peptides (Taylor, 1993). This suggests that protein degradation, or even direct methionine assimilation may provide additional sulfur sources to bacteria.

Taken together, the increasing trajectory of DMS:MeSH throughout the experiment reflects changes in bacterial sulfur

demand and the availability of other organosulfur molecules in the system. The pre-bloom stage of this experiment where DMS:MeSH was low and stable (4.60 ± 0.93) follows ambient conditions, such as those observed in Lawson et al. (2020) (Table 1), and the significantly higher (31.8 ± 18.7) ratio observed in the bloom and decay stage in this work are likely the product of an intense induced phytoplankton bloom and water mixing conditions not usually observed in the open ocean, but could be reflective of intense blooms in coastal environments.








**Table 1. DMS:MeSH measurements from this work and previous studies.**

| Measurement | DMS:MeSH | Notes | Reference |
|---|---|---|---|
| Airside | 4.60 ± 0.93 | Mesocosm experiment; Pre-bloom | This work |
| | 31.8 ± 18.7 | Mesocosm experiment; Bloom and decay | |
| Airside | 3–33 | Southwest Pacific Ocean | Lawson et al. (2020) |
| Flux | 3–7 | | |
| Waterside | 2–5.3 | Subarctic Northeast Pacific Ocean | Kiene et al. (2017) |
| Waterside | 1–30 | Atlantic Ocean | Kettle et al. (2001) |
| Waterside | 16.4 (mean) | Baltic Sea | Leck and Rodhe (1991) |
| | 19.7 (mean) | Kattegat-Skagerrak | |
| | 6.1 (mean) | North Sea | |


## 4 Conclusions and Outlook

During an induced phytoplankton bloom on coastal seawater, non-DMS organosulfur molecules accounted for on average 37% of the total gas-phase sulfur budget in the pre-bloom stage when chlorophyll-a was low, representative of ambient conditions in a typical coastal environment. The ratio of DMS:MeSH increased significantly during the phytoplankton bloom, likely due to the interaction between several variables influencing the molecules' production and loss processes in the seawater. DMS:MeSH was primarily sensitive to bacterial sulfur demand and the chemical composition and magnitude of available sulfur sources during the bloom. The low DMS:MeSH measured during the pre-bloom at the beginning of the experiment and which is more representative of average in situ conditions suggests MeSH can have a significant impact on atmospheric oxidative capacity and secondary sulfate formation in coastal environments given that it reacts with the hydroxyl radical seven times faster than DMS and has an expected unit yield of $SO_2$. This finding, combined with the significant emission of benzothiazole and substantial concentrations of other sulfur gases observed in this experiment, suggest pathways to secondary sulfate formation in a coastal environment warrant further study. A more complete understanding of coastal emissions of gaseous precursors to sulfate aerosol will improve model estimates of cloud formation and radiative balance in the marine environment.


**Data Availability:** Seawater measurements and Vocus PTR-ToF-MS measurements of DMS, MeSH, benzothiazole, and total other sulfur will be made available at https://minds.wisconsin.edu/handle/1793/76304.



**Supplement:** Wave channel and mesocosm details, additional methods descriptions, DMS:MeSH salinity discussion, and
supporting tables and figures.

**Author Contributions:** Conceptualization: DBK, GAN, JSS, KM, CL, CDC, AHG, KAP, THB; Investigation: DBK, GAN, JSS, ANM, JD, SA, EBB, MW, CKM, TP, FM, DRC, THB; Data Analysis: DBK, GAN, JSS, ANM, JD, SA, EBB, CKM, TP, FM, DRC, THB; Writing: DBK, JSS, JD, THB; All authors reviewed and edited the paper.


**Competing interests:** The authors declare that they have no conflict of interest.

**Acknowledgments:** We would like to thank the Center for Aerosol Impacts on Chemistry of the Environment (an NSF Chemical Innovation Center, (CHE-1801971)) and the full team which participated in the SeaSCAPE campaign for their
support of this work. We specifically thank Michael Alves for his contribution of the initial benzothiazole observation and identification at SeaSCAPE and related support of this publication.

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
