# Peer review of "Marine gas-phase sulfur emissions during an induced phytoplankton bloom"

_Atmospheric Chemistry and Physics, 2021_

## Author Comment (AC1)

Comments for all Reviewers: Thank you for taking the time to review this manuscript and for providing helpful and thoughtful comments. These suggestions have helped improve the quality of the manuscript. Below are responses (in green text) to the reviewer comments (in black text).

Response to Reviewer 1:

This manuscript describes CI-TOF-MS measurements of gaseous sulfur compounds emitted during a controlled, induced bloom experiment. The authors sought to determine if additional sulfur compounds, besides DMS, are emitted to the atmosphere, as well as the relative amounts. This is important for aerosol and cloud formation in the marine atmosphere. The manuscript reads well and the results are robust. The manuscript should be published after the following comments have been addressed.

Specific comments:

Lines 95-96: Missing current Lennartz references. For example (this alone might be good enough to reference):

Lennartz, S. T., Marandino, C. A., von Hobe, M., Andreae, M. O., Aranami, K., Atlas, E., Berkelhammer, M., Bingemer, H., Booge, D., Cutter, G., Cortes, P., Kremser, S., Law, C. S., Marriner, A., Simó, R., Quack, B., Uher, G., Xie, H., and Xu, X.: Marine carbonyl sulfide (OCS) and carbon disulfide (CS$_2$): a compilation of measurements in seawater and the marine boundary layer, Earth Syst. Sci. Data, 12, 591–609, https://doi.org/10.5194/essd-12-591-2020, 2020.

Authors' response: Thank you for bringing this to our attention. We have added the suggested Lennartz reference to line 93 of the revised manuscript.

Section 2.3: is there a reason why the authors chose not to use an equilibrator?

Authors' response: The authors thank the reviewer for this comment. Headspace equilibrators, such as those coupled to PTR-MS instruments like in Wohl et al. (2019) are a great way to assess dissolved VOC concentrations in seawater. For the SeaSCAPE experiment, multiple analytical goals made the usage of cryo purge and trap preferable, namely measurements of dissolved and particulate DMSP which required overnight reactions to yield measurable DMS. In addition, we sought to follow established protocols of dissolved DMS measurements, requiring close attention to water agitation which can disrupt phytoplankton cell membranes and can increase DMS concentrations significantly. For these reasons, purge and trap was chosen, however the suggestion to use an equilibrator is appreciated and will be strongly considered for future experimentation.

Figure 1b: are the individual peak shapes determined by injection of individual standards?

Authors' response: The individual peak shape for DMS was determined by a compressed gas standard. Other individual peak shapes in Figure 1b were determined by the processing software (Tofware v3.1.2, TOFWERK). Inputs into the software are a mass calibration, defined peak shape, width, and baseline, and a reference mass spectrum. Peaks are then found using a smoothed second-derivative approach and peak fits are applied.

Line 191: how were LODs determined?

Authors' response: Limits of detection were determined based on equation 1 in Bertram et al. (2011) for a signal-to-noise ratio of 3. This information has now been included in line 189-191, reading: "Limits of detection for DMS, MeSH, and benzothiazole at 1 minute averaging time were $0.20 \pm 0.49$ ppt, $1.5 \pm 0.25$ ppt, and $0.42 \pm 0.14$ ppt, respectively, calculated according to equation 1 in Bertram et al. (2011) for a signal-to-noise ratio of 3."

Line 201: why does MeSH have a water vapor dependence, but DMS does not? Have the authors considered an isotope dilution method to solve this sensitivity problem during the measurement?

Authors' response: We think that the water vapor dependence observed for MeSH and not observed for DMS could be, at least in part, due to conversion on inlet surfaces. This has been observed previously in MeSH measurements (Perraud et al., 2016). We have used isotope dilution methods in past field studies and agree this could be a useful method to confirm our calibrations from this experiment going forward.

Lines 212-220: Did the authors try using Teflon fittings? And Sulfinert® coated metals?

Authors' response: We used a mixture of both Teflon and stainless steel fittings connecting the ISV to the Vocus inlet during SeaSCAPE. Stainless steel was used on the ISV because the ISV was used for both gas and aerosol sampling. We recognize that for experiments only measuring gases, we would benefit from a fully Teflon inlet manifold. We did not try Sulfinert® coated metals in these experiments.

Line 284: I am not sure why this should be surprising if you don't have oxidation products – even if MSAM is outgassing, if no DMSO2 is formed then why would the authors expect a correlation?

Authors' response: The intention here was to suggest that given MSAM was a large emission in the Arabian Sea, especially relative to DMS, and was proposed to be formed from biogenic outgassing and not an oxidation product (Edtbauer et al., 2020), we were surprised to not see a

more significant MSAM emission. This suggests that MSAM was formed via a mechanism active in the Arabian Sea, but not in our study using coastal seawater from La Jolla, California.

We have changed the text from "Methane sulfonamide (MSAM) measured on average less than 1 ppt during the experiment and showed no positive correlation with the DMS oxidation product, $DMSO_2$ (Fig. S6). This is surprising as MSAM was recently measured at mixing ratios of up to 60 ppt, corresponding to up to 33% of DMS, in upwelling areas of the Arabian Sea and was correlated with $DMSO_2$ (r = 0.8) (Edtbauer et al., 2020)." to "Methane sulfonamide (MSAM) measured on average less than 1 ppt during the experiment. In contrast, Edtbauer et al. (2020) measured MSAM at mixing ratios of up to 60 ppt, corresponding to up to 33% of DMS, in upwelling areas of the Arabian Sea and suggested MSAM had a direct oceanic emission source. Results from our study suggest that the pathway producing MSAM in the Arabian Sea was not active in our experiments utilizing coastal seawater from Southern California."

Lines 304-306: this is only true if the atmospheric loss is the same or negligible. Is it true here that it is assumed to be negligible?

Authors' response: Since we are measuring from a low-oxidant headspace, continuously purged with zero air, we are assuming atmospheric loss is negligible. We have updated the language in lines 300-303 to clarify this, reading "Given that the Henry's law constants and diffusion constants for DMS and MeSH are roughly the same, and assuming atmospheric loss is negligible in the ISV, we expect the measured molar ratio of DMS:MeSH in the headspace to reflect the dissolved concentration ratio in the seawater."

Line 363: why would the authors assume the lifetimes to be the same?

Authors' response: Thank you for this note. You bring up a good point that the lifetimes of DMSP and DMS in the water are variable across the ocean, as shown in Table 1 of Vila-Costa et al., (2008). We have removed from line 363 the text reading "assuming the lifetimes of DMS and $DMSP_t$ in the water are the same" and instead have replaced it with our reasoning for using this $DMS:DMSP_t$ ratio. We have added the following text: "Further, since we do not have direct measurements of DMSP and DMS cycling, we will use the waterside $DMS:DMSP_t$ ratio to estimate the $DMSP_t$ to DMS conversion efficiency in seawater (Galí et al., 2018, 2021)."

Lines 366-370: how does this explanation account for bacterial cleavage?

Authors' response: As written, lines 366-370 do not explicitly account for bacterial cleavage. In this section, we propose that the low waterside $DMS:DMSP_t$ observed in this study is either a result of low DMS production from DMSP or a large DMS loss. Low DMS production from DMSP could be due to the phytoplankton community present in this study (diatom-dominant) or from reduced bacterial cleavage of DMSP to form DMS. Given the strong correlation between DMS:MeSH and methionine aminopeptidase activity (Fig. 5h), it is possible that protein

degradation provided the primary carbon and sulfur sources to bacteria and the DMSP to DMS
conversion pathway was not the most important metabolism.

Line 426: when I look at Figure 5, it does not seem like bacterial sulfur demand is the most
important determinant.

Authors' response: Thank you for this comment. While there is not as strong of a correlation
between DMS:MeSH and bacterial sulfur demand as there is between DMS:MeSH and other
variables plotted in Figure 5, we think that the temporal dynamics of bacterial sulfur demand in
relation to available sulfur (and the composition of available sulfur sources) is what makes
bacterial sulfur demand important for controlling DMS:MeSH. This can be seen in Fig. 6c where
there are two stages of the experiment separated by the vertical dashed line, where the high
DMS:MeSH values are observed in the latter stage when the estimated assimilated sulfur
exceeds the bacterial sulfur demand. We recognize how difficult it is to parse out these complex
relationships in large-scale experiments like this one and as such have changed the text in line
382 from "leading to a significant increase in measured DMS:MeSH" to "likely responsible for
part of the significant DMS:MeSH increase."

Supplemental material, paragraph starting at line 109: since this correlation looks so good, I
wonder if there is anything with which salinity correlates that could be interesting for the ratio
(but was not tested against the ratio).

Authors' response: None of the waterside variables measured (FDOM, chlorophyll-a, dissolved
oxygen, and water temperature) or other sulfur-containing ions correlate with DMS:MeSH above
$R^2 = 0.55$. The strongest correlation observed is between DMS:MeSH and day of the bloom ($R^2
= 0.64$). As discussed in the SI, the salinity decline throughout the bloom was likely a result of
additions of ultrapure water to the wave channel meant to maintain the water level, shown by the
correlation between DMS:MeSH and time. While the correlation between DMS:MeSH and
salinity is interesting, we do not think that the fluctuation in salinity was enough to cause the
changes in DMS:MeSH, meaning the observation is reflective of a correlation and not causation
(Salgado et al., 2014).

Response to Reviewer 2:

The paper entitled "Marine gas-phase sulphur emissions during an induced phytoplankton
bloom" by Kilgour and co-workers presents wind tunnel measurements of the emissions of
marine reduced sulphur species during phytoplankton blooms.

The paper is well written and discusses two important points.

Firstly, the authors report that during the pre-bloom period non-DMS species contribute significantly to the reduced sulphur budget. Methanethiol and benzothiazole were found to be the largest contributors to non-DMS sulphur emissions. The authors discuss the implications of this new finding for the marine sulphur budget, in particular the implications benzothiazole emission on new particle formation in marine environments.

Secondly the authors propose that the ratio DMS:MeSH is driven primarily by methionine aminopeptidase which catalyses cleavage of amino acids from proteins and peptides. The propose that salinity is not the main driving force for the DMS:MeSH ratio.

The paper falls within the scope of atmospheric chemistry and physics and is well written. I recommend the paper for acceptance subject to minor revisions.

Minor revisions suggested:

- The authors cite an unpublished work Franklin et al. 2021. I am not sure what the journal policy in this matter is, but it may be good if the paper can be deposited as a preprint in a repository so that it becomes accessible to the reader.

Authors' response: Franklin et al. (2021) has been accepted in the journal Environmental Science and Technology. We have updated the reference with the following doi: 10.1021/acs.est.1c04422

- The authors also cite and AGU abstract, which is not a peer reviewed source to support one of their points Kiene et al. 2017. This is not ideal. Have the authors checked whether the relevant data has been published since, possible under a different title or is available in some repository.

Authors' response: Thank you for noting this. We are not aware of where this data has been published and peer-reviewed and as a result, we have removed references to it in the manuscript.

- The Sander reference for Henry's law constants appears to be incomplete Atmos. Chem. Phys., 15, 4399–4981, 2015 doi:10.5194/acp-15-4399-2015

Authors' response: Thank you for catching this mistake. We have updated the reference to accurately cite this paper.

**References**

Bertram, T. H., Kimmel, J. R., Crisp, T. A., Ryder, O. S., Yatavelli, R. L. N., Thornton, J. A., Cubison, M. J., Gonin, M., and Worsnop, D. R.: A field-deployable, chemical ionization time-of-flight mass spectrometer, Atmospheric Meas. Tech., 4, 1471–1479, https://doi.org/10.5194/amt-4-1471-2011, 2011.

Edtbauer, A., Stönner, C., Pfannerstill, E. Y., Berasategui, M., Walter, D., Crowley, J. N., Lelieveld, J., and Williams, J.: A new marine biogenic emission: methane sulfonamide (MSAM), dimethyl sulfide (DMS), and dimethyl sulfone (DMSO$_2$) measured in air over the Arabian Sea, Atmospheric Chem. Phys., 20, 6081–6094, https://doi.org/10.5194/acp-20-6081-2020, 2020.

Galí, M., Levasseur, M., Devred, E., Simó, R., and Babin, M.: Sea-surface dimethylsulfide (DMS) concentration from satellite data at global and regional scales, Biogeosciences, 15, 3497–3519, https://doi.org/10.5194/bg-15-3497-2018, 2018.

Galí, M., Lizotte, M., Kieber, D. J., Randelhoff, A., Hussherr, R., Xue, L., Dinasquet, J., Babin, M., Rehm, E., and Levasseur, M.: DMS emissions from the Arctic marginal ice zone, Elem. Sci. Anthr., 9, 00113, https://doi.org/10.1525/elementa.2020.00113, 2021.

Perraud, V., Meinardi, S., Blake, D. R., and Finlayson-Pitts, B. J.: Challenges associated with the sampling and analysis of organosulfur compounds in air using real-time PTR-ToF-MS and offline GC-FID, Atmospheric Meas. Tech., 9, 1325–1340, https://doi.org/10.5194/amt-9-1325-2016, 2016.

Salgado, P., Kiene, R., Wiebe, W., and Magalhães, C.: Salinity as a regulator of DMSP degradation in Ruegeria pomeroyi DSS-3, J. Microbiol., 52, 948–954, https://doi.org/10.1007/s12275-014-4409-1, 2014.

Vila-Costa, M., Kiene, R. P., and Simí, R.: Seasonal variability of the dynamics of dimethylated sulfur compounds in a coastal northwest Mediterranean site, Limnol. Oceanogr., 53, 198–211, https://doi.org/10.4319/lo.2008.53.1.0198, 2008.

Wohl, C., Capelle, D., Jones, A., Sturges, W. T., Nightingale, P. D., Else, B. G. T., and Yang, M.: Segmented flow coil equilibrator coupled to a proton-transfer-reaction mass spectrometer for measurements of a broad range of volatile organic compounds in seawater, Ocean Sci., 15, 925–940, https://doi.org/10.5194/os-15-925-2019, 2019.